

# PhosVarDeep: deep-learning based prediction of phospho-variants using sequence information

Xia Liu, Minghui Wang and Ao Li

School of Information Science and Technology, University of Science and Technology of China, Heifei, China

## ABSTRACT

Human DNA sequencing has revealed numerous single nucleotide variants associated with complex diseases. Researchers have shown that these variants have potential effects on protein function, one of which is to disrupt protein phosphorylation. Based on conventional machine learning algorithms, several computational methods for predicting phospho-variants have been developed, but their performance still leaves considerable room for improvement. In recent years, deep learning has been successfully applied in biological sequence analysis with its efficient sequence pattern learning ability, which provides a powerful tool for improving phospho-variant prediction based on protein sequence information. In the study, we present PhosVarDeep, a novel unified deep-learning framework for phospho-variant prediction. PhosVarDeep takes reference and variant sequences as inputs and adopts a Siamese-like CNN architecture containing two identical subnetworks and a prediction module. In each subnetwork, general phosphorylation sequence features are extracted by a pre-trained sequence feature encoding network and then fed into a CNN module for capturing variant-aware phosphorylation sequence features. After that, a prediction module is introduced to integrate the outputs of the two subnetworks and generate the prediction results of phospho-variants. Comprehensive experimental results on phospho-variant data demonstrates that our method significantly improves the prediction performance of phospho-variants and compares favorably with existing conventional machine learning methods.

## INTRODUCTION

Nowadays, human DNA sequencing studies have revealed millions of single nucleotide variants, which have been shown to significantly associate with complex diseases such as cancer and cardiovascular diseases (*Gonzalez-Perez et al., 2013*; *MacArthur et al., 2014*). Although millions of variants have been discovered, their exact effects on resultant RNA or protein products generally remain unknown (*Patrick et al., 2017*). One of the potential effects of these variants on protein function is to disrupt post-translational modifications especially protein phosphorylation (*Kim et al., 2015*; *Krassowski et al., 2018*), since phosphorylation is the most ubiquitous post-translational modification and plays an important role in understanding the alteration of signaling pathways caused by variations

Corresponding author
Minghui Wang, mhwang@ustc.edu.cn

(*Reimand & Bader, 2013*; *Reimand, Wagih & Bader, 2013*). Therefore, identifying and understanding variants that affect phosphorylation status is critical in the study of cell biology, disease treatment and prevention. Here, we follow the previous studies (*Ryu et al., 2009*; *Patrick et al., 2017*) to use the term 'phospho-variant' to refer to a variant that impacts the phosphorylation status of an amino acid. The examples of phospho-variants used in this paper include those variants that modify the $S/T/Y$ residue or adjacent residue, *i.e.,* Type I, Type II and Type III defined in *Ryu et al. (2009)*.

Indeed, it has been reported that there are numerous phospho-variants that are capable of impacting protein phosphorylation. For example, over 19,000 missense mutations are included in the PhosphoSitePlus PTMVar dataset (*Hornbeck et al., 2015*), which fall within a 15-residue window centered by an experimentally-identified phosphorylation site and may preeminently disrupt existing phosphorylation sites or introduce new phosphorylation sites. Meanwhile, several databases have also been developed to catalogue the suspected effects of variants on potential phosphorylation sites. For example, *Ryu et al. (2009)* search for known phospho-variants and predict other possible phospho-variants among human variations by PredPhospho, which are then incorporated into the Phospho-variant database. Subsequently, by string matching with 23,978 phosphorylation sites on human, *Ren et al. (2010)* detect potential phospho-variants and compile them into the PhosSNP database.

In contrast to the above approaches providing a database, there are several computational methods based on conventional machine learning algorithms for detecting and analyzing phospho-variants. For example, *Wagih, Reimand & Bader (2015)* developed a Bayesian statistics-based method called mutation impact on phosphorylation (MIMP), which constructs position weight matrices and trains Gaussian mixture models to predict the function of variants on phosphorylation sites. Subsequently, established on the previous Bayesian network model for phosphorylation site prediction, *Patrick et al. (2017)* presented an effective method called PhosphoPICK-SNP to quantify the expected impacts of variants on protein phosphorylation status. The PhosphoPICK-SNP method obtains prediction scores from a pair of reference and variant protein sequences surrounding a potential phosphorylation site containing missense mutation, and then combines them to analyze the impacts of variation on protein phosphorylation. In this way, *Patrick et al. (2017)* predict the effects of known phospho-variants on phosphorylation and construct a background distribution of proteome-wide predicted variant effects to detect novel examples of phospho-variants.

Recently, as a rising and promising machine learning technique, deep learning has made a remarkable breakthrough in many areas such as image recognition (*Rawat & Wang, 2017*) and natural language understanding (*Collobert et al., 2011*). Compared with conventional machine learning techniques, deep learning methods have a distinctive advantage that allows automatically discovery of the complex representations needed for downstream task. Among them, convolutional neural network (CNN) (*Krizhevsky, Sutskever & Hinton, 2017*) has been successfully undertaken in biological sequence analysis for its powerful capability of learning sequence patterns. For example, *Alipanahi et al. (2015)* employed CNN in DeepBind to predict sequence specificities of DNA- and RNA-binding proteins.
In addition, for phosphorylation site prediction, *Wang et al. (2017)* proposed Musitedeep based on a multi-layer CNN architecture with attention mechanism, and lately *Luo et al. (2019)* presented DeepPhos using densely connected CNN architectures to learn multiple representations of protein sequences. These deep learning methods using CNN architectures have obtained better performance than conventional machine learning methods. However, so far there is no approach to address the problem of phospho-variant prediction by deep learning and it is nontrivial to develop such a powerful tool that can effectively utilize both reference and variant sequence information to predict impacts of variants on protein phosphorylation status.

In this work, we propose PhosVarDeep, a novel unified deep-learning framework for phospho-variant prediction by efficiently extracting and combining both reference and variant protein sequential information. PhosVarDeep employs a Siamese-like CNN architecture containing two identical subnetworks with shared weights. Each subnetwork is composed of a sequence feature encoding network (PhosFEN) and a multi-layer CNN (CNN module). To begin, we utilize PhosFEN to capture general phosphorylation sequence features of the reference and variant sequences, which are fed into the CNN module to further learn variant-aware phosphorylation sequence features jointly. Then we employ a prediction module to combine the two obtained features from the subnetworks to produce a prediction score that best separates the positive and negative examples of phospho-variants. We conduct comprehensive experiments to study the performance of PhosVarDeep, and the evaluation results exhibit that our proposed method significantly improves the predictive performance in identifying phospho-variants and is superior to existing prediction methods.

## MATERIALS & METHODS

### Dataset and pre-process

Here, in order to train and evaluate our method, we collect over 2,000 potential phospho-variants in PhosSNP (*Ren et al., 2010*) as the positive set and adopt three negative sets generated by *Patrick et al. (2017)*, which contain information of protein UniProt index, amino acid variation and phosphorylation sites. The positive dataset in PhosSNP we adopt is based on exact string matching reference/variant sequence with experimentally identified human phosphorylation sites. Accordingly, either identical hit in reference or variant sequence is considered as potential phospho-variants (*Ren et al., 2010*). As for negative sets, as described in Ralph et al., the three high confidence negative sets adopted in this study are generated by different criteria under the fact that phosphorylation sites are less likely to occur (1) in solvent inaccessible/buried regions of a protein; (2) in transmembrane domains; (3) in proteins that do not interact either directly or through mediators with a kinase (*Patrick et al., 2017*). Next, we map the collected phospho-variant data with protein sequences by its corresponding UniProt index and amino acid variation so that each phospho-variant corresponds to a pair of reference and variant protein sequences (*Bateman et al., 2015*). Table 1 shows the sizes of the positive set and three negative sets used for phospho-variant prediction, with respect to different phosphorylation site types,

**Table 1** The sizes of positive and negative sets with respect to different phosphorylation site types.

|  | S/T sites | Y sites |
|---|---|---|
| Positive set | 763 | 440 |
| Negative set1 | 5796 | 2372 |
| Negative set2 | 2285 | 715 |
| Negative set3 | 17204 | 9901 |

*i.e.,* serine (S)/threonine (T) or tyrosine (Y). Given a phospho-variant, we intercept the protein fragments centered at a phosphorylation site on its corresponding pair of reference and variant sequences. Then each protein fragment is encoded by the one-hot encoding strategy into a $L \times N$ two-dimension matrix, where $L$ represents the window size of the protein fragment, and N is set to 21 according to the total number of common amino acids (*Min et al., 2017*). Besides, we apply CD-HIT with similarity threshold of 40% to all the collected data to reduce the sequence redundancy by following previous studies (*Pan et al., 2014*; *Zhao et al., 2012*).

After completing the data pre-processing, the positive set is combined with each negative set to form three phospho-variant datasets, and for each phospho-variant dataset, we train two deep learning models using phospho-variant data on S/T and Y sites respectively. Meanwhile, we use a performance evaluation strategy commonly adopted in deep learning methods for sequential analysis (*Zhou & Troyanskaya, 2015*; *Khurana et al., 2018*), that is, each dataset is randomly divided into strictly non-overlapping training, validation and test sets and the ratio is set to 6:2:2 in our study. In this way, we adopt the training data to tune the model weights and utilize the validation data to prevent overfitting (*Zhou & Troyanskaya, 2015*). The test set is adopted to evaluate the performance of PhosVarDeep and to implement the comparison of PhosVarDeep and other phospho-variant prediction approaches.

## Siamese-like architecture of PhosVarDeep

In this study, we design a Siamese-like deep-learning framework for phospho-variant prediction. As a classic metric learning method, Siamese neural network is first introduced in *Bromley et al. (1993)* and has been successfully adopted in tasks such as signature verification (*Chopra, Hadsell & LeCun, 2005*), face verification (*Cao, Ying & Li, 2013*) and object tracking (*Bertinetto et al., 2016*). To measure similarity between pairs of inputs, Siamese neural network learns a function mapping input patterns to the representation or target space where similar pairs will get close and non-similar pairs will get away from each other (*Zagoruyko & Komodakis, 2015*). Generally, Siamese neural network contains two subnetworks sharing same configuration, weights, and parameters, which ensures that two similar inputs are transformed into similar feature representations. Here, we adopt a deep-learning framework in Siamese style to learn a complex non-liner mapping for separating the positive and negative samples of the phospho-variants. Figure 1 shows the proposed architecture of PhosVarDeep, which consists of two identical subnetworks with shared weights and a prediction module. Specifically, within each subnetwork, the PhosFEN and the CNN module extract a high dimensional feature representation of an

 

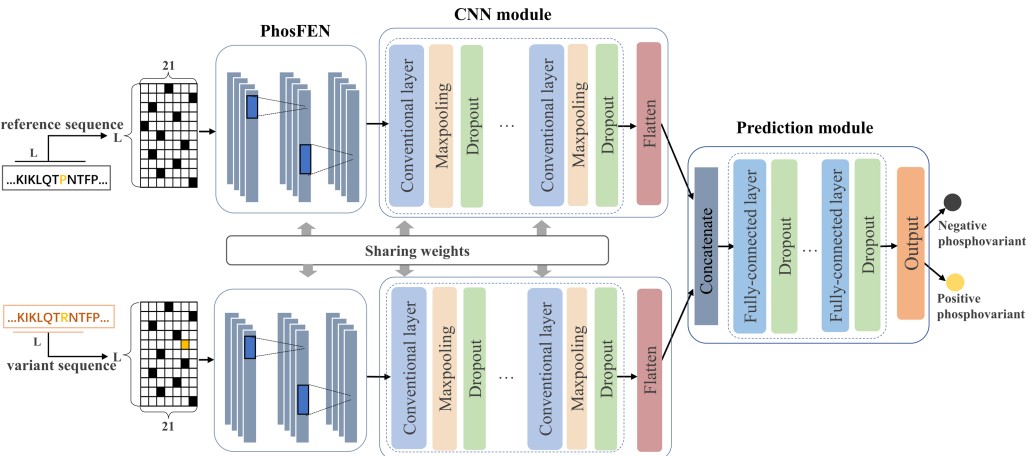

**Figure 1** **Illustration of the proposed PhosVarDeep framework.**

input sequence. Next, the prediction module is introduced to integrate the outputs of the two subnetworks as combined features to generate prediction results of phospho-variants. The details of those networks are described as follows.

## PhosFEN

In each subnetwork, PhosFEN presents a sequence feature encoding network to extract the respective features from the reference and variant local sequences of a given phospho-variant. Specifically, the input of PhosFEN is one-hot local sequence $e_s \in R^{L \times 21}$ ($s = s_1, s_2$), with $s_1$ and $s_2$ being the reference and variant sequences, respectively. Then the sequence features related to phosphorylation are extracted by PhosFEN as follows:

$$f_s = \text{PhosFEN}\left(e_s; W^F\right) \tag{1}$$

where $W^F$ represents all the parameter matrices and bias items in PhosFEN. In this way, the feature map $f_s$ is exported from PhosFEN as general phosphorylation-related local features, and the pair of $f_{s_1}$ and $f_{s_2}$ is used as input to the following CNN module.

In this work, since the amount of training data of phospho-variants is much smaller than the size of parameters in deep learning models, overfitting is likely to occur in the training process. In order to solve the problem of insufficient training data, we have studied the feature learning capabilities of the existing deep-learning models for phosphorylation site prediction and adopt DeepPhos (*Luo et al., 2019*) in our study through transfer learning (*Yosinski et al., 2014*). DeepPhos is a pre-trained deep learning model consisting of densely connected CNN blocks, which can capture the complex deep sequence representations relevant to phosphorylation better than other deep-learning models. We transfer the whole layers of the model to PhosFEN except its final three layers including the flatten layer, the fully connected layer, and the output layer in our study.

## CNN module

After the general phosphorylation sequence features are extracted from PhosFEN, they are fed into the CNN module to generate variant-aware phosphorylation sequence features.

**Table 2   Details of CNN module and prediction module.**

|  | Layers | Details |
|---|---|---|
| Multi-layer CNN | Conventional layer(+ReLU) | 32 filters |
|  | Conventional layer(+ReLU) | 64 filters |
|  | Conventional layer(+ReLU) | 128 filters |
|  | Dropout | $P = 0.3$ |
|  | MaxPooling | pool_size $= 2$ |
| Multi-layer DNN | Fully connected layer(+ReLU) | 128 neurons |
|  | Fully connected layer(+ReLU) | 64 neurons |
|  | Fully connected layer(+ReLU) | 32 neurons |
|  | Dropout | $P = 0.3$ |
| Output layer | Fully connected layer (+softmax) | 2 neurons |

As shown in Fig. 1, each CNN module is comprised of multiple convolutional layers and max pooling layers, and a flatten layer. The convolutional layer generates a feature map by convoluting the input with a set of convolution kernels. Mathematically, for input phosphorylation-related features $f_s$ $(s = s_1, s_2)$, let $h_{i,s}$ be the ith feature map in output, and the convolutional layers can be described as follows:

$$h_{1,s} = \alpha \left( W_1^k f_s + b_1^k \right) \tag{2}$$

$$h_{i,s} = \alpha \left( W_i^k h_{i-1,s} + b_i^k \right) 2 \leq i \leq M \tag{3}$$

where $W_i^k$ and $b_i^k$ refer to the parameter matrices and bias items in the $i$th convolutional layer, with $k$ being the number of convolutional kernels. $\alpha$ represents ReLU activation function that can realize the nonlinear transformation, $M$ refers to the number of convolutional layers and here is set to 3. In this way, the variant-aware phosphorylation sequence features can be generated, and then by the flatten layer, they are transformed into a pair of one-dimensional tensors $(h_{s_1}, h_{s_2}) \in R^d$. In addition, to relieve the risk of overfitting during the training process, a dropout layer is added after each convolutional layer to discard some neurons randomly (*Srivastava et al., 2014*). The details of the CNN Module are listed in Table 2.

## Prediction module

Taking a pair of sequence features $h_{s_1}$ and $h_{s_2}$ as inputs, the prediction module utilizes a multi-layer DNN consisting of three fully connected layers to integrate the feature pair and compute a prediction score for phospho-variant. In detail, the two input features are concatenated and then fed into the multi-layer DNN to capture the abstract combined features $c_v \in R^u$, here $u$ refers to the number of neurons in the final fully connected layer. Next, the output layer with softmax as the activation function is used to generate prediction scores of the positive and negative phospho-variant, which can be calculated as follows:

$$P \left( y = 1 | (s_1, s_2) \right) = \frac{1}{1 + \exp \left( -c_v W_v \right)} \tag{4}$$

$$P(y = 0|(s_1, s_2)) = 1 - P(y = 1|(s_1, s_2))$$ (5)

where $W_v \in R^{u \times 2}$ represents the weight matrix of softmax function. The details of the multi-layer DNN and output layer are listed in Table 2.

## Training

PhosVarDeep is a unified deep learning framework for phospho-variant prediction and is trained to classify the phospho-variants into two classes: positive phospho-variants and negative phospho-variants. Accordingly, for the aim of minimizing training error, here we adopt a binary cross-entropy as loss function:

$$L_c = -\frac{1}{N} \sum_{j=1}^{n} y^j \ln P(y^j = 1|(s_1, s_2)^j) + (1 - y^j) \ln P(y^j = 0|(s_1, s_2)^j)$$ (6)

where N refers to the size of training data, $(s_1, s_2)^j$ represents the pair of reference and variant sequences for the $j$ th input phospho-variant and $y^j$ refers to the corresponding class label. We freeze all the layers of PhosFEN and train the CNN module and prediction module jointly on phospho-variant training data, in which the weights and biases of conventional layers and fully connected layers are the parameters to be estimated. Besides, as a widely used stochastic gradient descent algorithm, Adam optimizer (*Kingma & Ba, 2015*). is adopted in the training process. At the same time, we use mini-batch training strategy in this study to randomly divide small proportions of the training samples in each iteration into optimizer loops. In addition, to deal with the problem of data imbalance, we follow the previous study (*Wang et al., 2017*) to apply a bootstrapping strategy in our deep learning method.

## Performance assessment

In this study, several widely used measurements are leveraged to assess the performance of our proposed PhosVarDeep, which include specificity (Sp), sensitivity (Sn), precision (Pre), overall accuracy (Acc), F1 scores and Matthew's correlation coefficient (MCC). Their detailed definitions are as below:

$$Sp = \frac{TN}{TN + FP}$$ (7)

$$Sn = \frac{TP}{TP + FN}$$ (8)

$$Pre = \frac{TP}{TP + FP}$$ (9)

$$Acc = \frac{TP + TN}{TP + TN + FP + FN}$$ (10)

$$F1 = \frac{2 \times Pre \times Sn}{Pre + Sn} \tag{11}$$

$$MCC = \frac{TP \times TN - FP \times FN}{\sqrt{((TP + FN) \times (TP + FP) \times (TN + FN) \times (TN + FP))}} \tag{12}$$

where TP, FP, TN, and FN denote true positives, false positives, true negatives, and false negatives, respectively. The other measurements are calculated based on TP, FP, TN, and FN. Moreover, we plot the receiver operating characteristic curve (ROC) and calculate the area under ROC curve (AUC) to evaluate the overall performance.

## EXPERIMENTS & RESULTS

### Determining the PhosFEN model

To begin, in order to extract general phosphorylation sequence features, we transfer phosphorylation prediction models to PhosFEN in our method. There are several pre-trained models with good phosphorylation-related sequence feature learning capabilities. Among them, Musitedeep (*Wang et al., 2017*) and DeepPhos (*Luo et al., 2019*) are both CNN models achieving significantly better performance than previous methods. To determine which model provides more useful information for phospho-variant prediction, we compare the two phosphorylation site predictors using an existing comprehensive phosphorylation site dataset (*Luo et al., 2019*) collected from Phospho.ELM, PhosphositePlus, HPRD, dbPTM and SysPTM. The Roc curves and AUC values of the two models are shown in Fig. 2. It is obvious that DeepPhos consistently achieved higher performance on both S/T sites and Y sites than Musitedeep. For S/T sites, the AUC value of DeepPhos is 3.0% higher than that of Musitedeep. For Y sites, compared with Musitedeep, the AUC value obtained by DeepPhos is increased by 5.7%. The above results suggest that DeepPhos has better phosphorylation-related feature learning capability on phospho-variant data. Accordingly, for accurate phospho-variant prediction, we employ pre-trained DeepPhos in PhosFEN to capture general phosphorylation sequence features in phospho-variant data.

### Performance evaluation of PhosVarDeep

In this part, to evaluate the performance of PhosVarDeep in extracting and integrating reference and variant sequence information for phospho-variant prediction, we conduct an ablation study by comparing three different model configurations: (1) PhosFEN*: in this case, the paired outputs of PhosFEN are directly combined and fed into a fully connected layer to get the prediction results; (2) PM*: in this case, we employ the prediction module to integrate the outputs of PhosFEN to predict phospho-variant; (3) PhosVarDeep.: in this case, for the paired outputs of PhosFEN, we utilize the CNN module to further learn variant-aware phosphorylation sequence features and employ the prediction module to generate combined features for final prediction.

We apply all above methods on three phospho-variant test sets and Table 3 lists their AUC values on S/T and Y sites. It is obvious that PM* produces higher performance
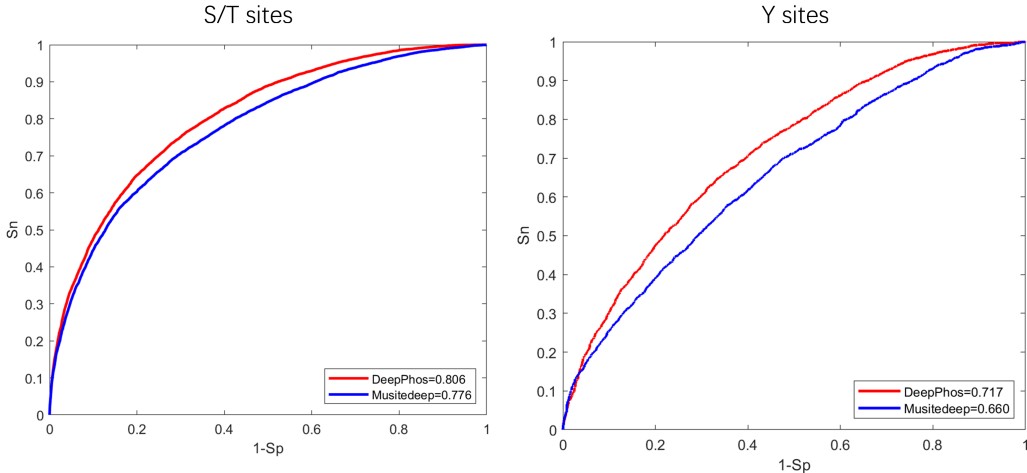

**Figure 2** ROC curves and AUC values of DeepPhos and Musitedeep for general phosphorylation site prediction on S/T and Y sites.

**Table 3** AUC values of PhosVarDeep for phospho-variant prediction.

| Method | Test set1 | | Test set2 | | Test set3 | |
|---|---|---|---|---|---|---|
| | S/T sites | Y sites | S/T sites | Y sites | S/T sites | Y sites |
| PhosFEN* | 0.845 | 0.827 | 0.915 | 0.898 | 0.719 | 0.661 |
| PM* | 0.909 | 0.878 | 0.930 | 0.917 | 0.848 | 0.812 |
| PhosVarDeep | **0.946** | **0.919** | **0.957** | **0.923** | **0.906** | **0.874** |

**Notes.**
Best performance values are highlighted in bold.

than PhosFEN*, which indicates the prediction module effectively contributes to the performance of phospho-variant prediction. For example, on test set2 the AUC values of PM* are 0.930 and 0.917 on S/T and Y sites respectively, which yield 1.5% and 1.9% improvement over PhosFEN*. More importantly, it can also be clearly observed that PhosVarDeep achieves better performance than PM* on all the test sets. For example, on test set3 the AUC values obtained by PhosVarDeep on S/T and Y sites reach 0.906 and 0.874 respectively, which are 5.8% and 6.2% higher than the corresponding AUC values of PM*. These results demonstrate that the CNN module can effectively learn variant-aware phosphorylation sequence features that are useful for predicting phospho-variant. At the same time, by integrating the CNN module and prediction module, PhosVarDeep consistently outperforms PhosFEN* with remarkable improvements on AUC value. For example, on test set1, the AUC value is enhanced from 0.845 (PhosFEN*) to 0.946 (PhosVarDeep) on S/T sites, and similarly, on Y sites the AUC value is increased from 0.827 to 0.919.

In addition to AUC values, we also utilize Sp, Sn, Acc, Pre, F1 and MCC to verify the effectiveness of the proposed method. Similar to previous studies (*Luo et al., 2019*; *Wang et al., 2021*), we calculate the values of other measurements when Sp is equal to high

**Table 4  The values (%) of Sn, Acc, MCC, Pre and F1 of PhosVarDeep on S/T sites.**

| | Method | Sp = 90% | | | | | Sp = 95% | | | | |
|---|---|---|---|---|---|---|---|---|---|---|---|
| | | Sn | Acc | Mcc | Pre | F1 | Sn | Acc | Mcc | Pre | F1 |
| Test set1 | PhosFEN* | 78.3 | 84.2 | 68.9 | 88.8 | 83.2 | 58.6 | 76.6 | 57.2 | 91.8 | 71.5 |
| | PM* | 83.6 | 86.8 | 73.8 | 89.4 | 86.4 | 70.4 | 82.6 | 67.2 | 93.0 | 80.1 |
| | PhosVarDeep | **92.1** | **91.1** | **82.3** | **90.3** | **91.2** | **80.9** | **87.8** | **76.4** | **93.9** | **86.9** |
| Test set2 | PhosFEN* | 81.6 | 85.9 | 72.0 | 89.2 | 85.2 | 78.3 | 86.5 | 74.0 | 93.7 | 85.3 |
| | PM* | 88.8 | 89.5 | 79.0 | 90.0 | 89.4 | 84.2 | 89.5 | 79.4 | 94.1 | 88.9 |
| | PhosVarDeep | **94.7** | **92.4** | **85.0** | **90.6** | **92.6** | **88.8** | **91.8** | **83.7** | **94.4** | **91.5** |
| Test set3 | PhosFEN* | 54.6 | 72.4 | 47.9 | 84.7 | 66.4 | 42.1 | 68.4 | 43.3 | 88.9 | 57.1 |
| | PM* | 65.1 | 77.6 | 57.1 | 86.8 | 74.4 | 47.4 | 71.1 | 47.8 | 90.0 | 62.1 |
| | PhosVarDeep | **71.1** | **80.6** | **62.3** | **87.8** | **78.5** | **61.2** | **78.0** | **59.4** | **92.1** | **73.5** |

**Table 5  The values (%) of Sn, Acc, MCC, Pre and F1 of PhosVarDeep on Y sites.**

| | Method | Sp = 90% | | | | | Sp = 95% | | | | |
|---|---|---|---|---|---|---|---|---|---|---|---|
| | | Sn | Acc | Mcc | Pre | F1 | Sn | Acc | Mcc | Pre | F1 |
| Test set1 | PhosFEN* | 60.2 | 75.0 | 52.3 | 85.5 | 70.7 | 56.8 | 76.1 | 56.7 | 92.6 | 70.4 |
| | PM* | 70.5 | 80.1 | 61.4 | 87.3 | 78.0 | 67.0 | 81.3 | 65.2 | 93.7 | 78.1 |
| | PhosVarDeep | **88.6** | **89.2** | **78.4** | **89.7** | **89.1** | **79.5** | **87.5** | **76.0** | **94.6** | **86.4** |
| Test set2 | PhosFEN* | 71.6 | 80.7 | 62.4 | 87.5 | 78.8 | 64.8 | 80.1 | 63.3 | 93.4 | 76.5 |
| | PM* | 81.8 | 86.4 | 73.0 | 90.0 | 85.7 | 71.6 | 83.5 | 69.0 | 94.0 | 81.3 |
| | PhosVarDeep | **90.9** | **90.3** | **80.7** | **89.9** | **90.4** | **87.5** | **91.5** | **83.2** | **95.1** | **91.1** |
| Test set3 | PhosFEN* | 52.3 | 71.0 | 45.4 | 83.6 | 64.3 | 15.9 | 55.7 | 18.8 | 77.8 | 26.4 |
| | PM* | 58.0 | 73.9 | 50.3 | 85.0 | 68.9 | 27.3 | 61.4 | 31.1 | 85.7 | 41.4 |
| | PhosVarDeep | **61.4** | **75.6** | **53.3** | **85.7** | **71.5** | **42.0** | **68.8** | **44.4** | **90.2** | **57.4** |

stringency level (95.0%) or medium stringency level (90.0%), respectively. The detailed measurement values on S/T and Y sites are separately displayed in Tables 4 and 5. From these results, we can see that the performance of PM* is better than PhosFEN* across all the test sets. Take test set2 on S/T sites as an example, when Sp is set at 90.0%, the Sn, Acc, MCC, Pre and F1 values of PM* are 7.2%, 3.6%, 7.0%, 0.8% and 4.2% higher than PhosFEN*, respectively. Also, Tables 4 and 5 clearly show the superior performance of PhosVarDeep. For example, on test set1 PhosVarDeep manages to obtain F1 scores of 0.912 at middle stringency level on S/T sites, with an improvement of 8.0% and 4.8% compared with PhosFEN* and PM*, respectively. On test set3 the Acc values of PhosVarDeep at high stringency level are increased by 13.1% and 7.4% on Y sites, respectively. The above results further verify that our proposed PhosVarDeep can effectively extract and combine the information of reference and variant sequences, which leads to a significant improvement in the performance of phospho-variant prediction.

## Comparison with existing methods

To further assess the performance of PhosVarDeep for phospho-variant prediction, we compare it against MIMP (*Wagih, Reimand & Bader, 2015*), a sequence-based method

**Table 6 AUC values of different methods for phospho-variant prediction.**

| Method | Test set1 | | Test set2 | | Test set3 | |
|---|---|---|---|---|---|---|
| | S/T sites | Y sites | S/T sites | Y sites | S/T sites | Y sites |
| MIMP | 0.797 | 0.725 | 0.830 | 0.737 | 0.739 | 0.611 |
| PhosphoPICK-SNP | 0.852 | 0.794 | 0.850 | 0.827 | 0.823 | 0.784 |
| PhosVarDeep | **0.945** | **0.914** | **0.956** | **0.922** | **0.902** | **0.871** |

and PhosphoPICK-SNP (*Patrick et al., 2017*), a motif analysis and context-based method. It is of note that we follow (*Patrick et al., 2017*) to perform the comparison using those phospho-variants for which we can obtain prediction results from MIMP.

As shown in Table 6, PhosVarDeep obtains the best performance on AUC values in comparison with other methods. Take test set2 as an example, compared with MIMP and PhosphoPICK-SNP, on S/T sites PhosVarDeep obtains 12.6% and 10.6% improvement for AUC values, respectively. Also, on Y sites, the AUC value achieved by PhosVarDeep is 0.922, which has an improvement of 9.5% over the next-best method. Furthermore, we compute the values of Sn, Acc, Pre, MCC and F1 for all the methods on the test sets on S/T sites, and the results are shown in Fig. 3. It can be observed that PhosVarDeep consistently performs better on all the metrics than MIMP and PhosphoPICK-SNP. For example, on test set3 at high stringency level, PhosVarDeep obtains Sn of 0.627, Acc of 0.790, MCC of 0.612, Pre of 0.927 and F1 of 0.748 on S/T sites, while the corresponding values of the next-best method are 0.310, 0.633, 0.341, 0.863 and 0.456, respectively. In addition to performing comparison on the three phospho-variant test sets, we adopt a list of experimentally confirmed phospho-variants compiled by *Patrick et al. (2017)* and there are 20 examples after deleting the overlapping data in the training set. The scores predicted by PhosVarDeep and MIMP are shown in Table 7. The closer the prediction score is to 1.0, the more likely it is to change the phosphorylation state. From Table 7, we can see that PhosVarDeep remarkably outperforms MIMP for predicting the experimentally confirmed positive examples of phospho-variants. To sum up, these results indicate that PhosVarDeep is a highly competitive and efficient method in predicting phospho-variant based on sequence information.

## Visualization of learned features

The combined features extracted by PhosVarDeep and the original combined one-hot encoding features of reference and variant sequences are visualized in this section to intuitively shows the ability of our proposed deep learning method in phospho-variant prediction. Here, we use a popular visualization algorithm t-SNE (*Van Der Maaten & Hinton, 2008*) and observe the difference between positive and negative examples of phospho-variants. Take test set3 as an example, the original combined sequence features and the abstract combined features extracted by our model are shown in Fig. 4. It suggests that we can hardly separate the positive examples of phospho-variant from negatives by original combined features, while as the extracted combined features show separate trends, we can distinguish these two classes more clearly by the deep representations of
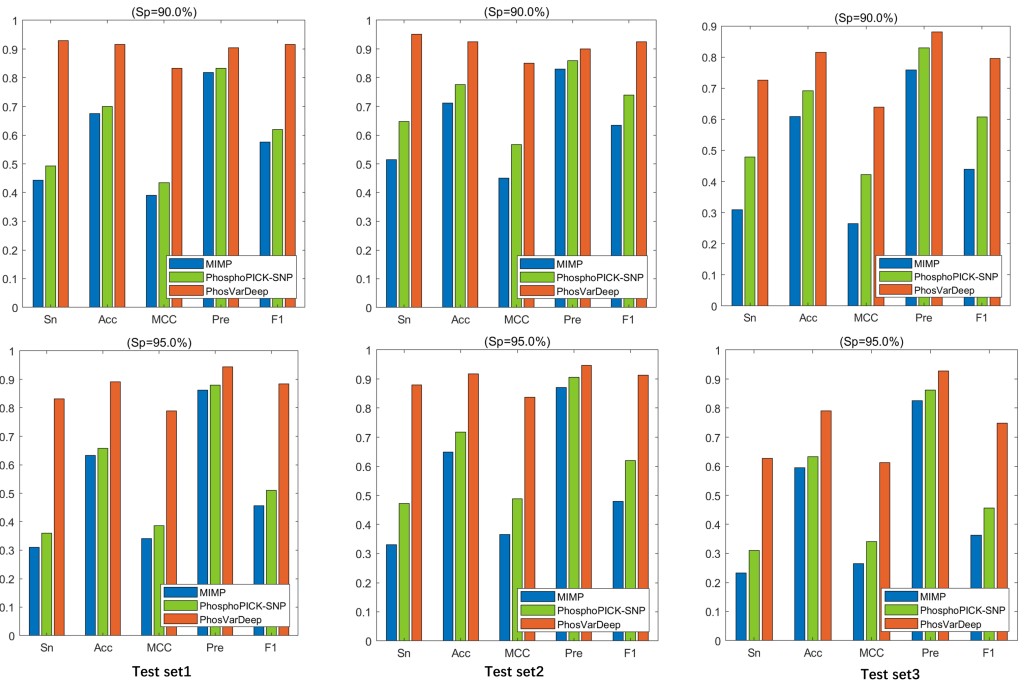

**Figure 3** The values of Sn, Acc, Pre and F1 of different methods at Sp = 90.0% and Sp = 95.0% on S/T sites.

PhosVarDeep. These results demonstrate that original reference and variant sequences can be combined and transformed into meaningful representations with stronger discriminant power by PhosVarDeep, which can be helpful for further analysis of phospho-variant prediction.

## DISCUSSION

In this study, we present PhosVarDeep, a unified deep-learning framework based on sequence information for accurate phospho-variant prediction. The experimental results demonstrate that PhosVarDeep obtains better performance than existing phospho-variant prediction methods evaluated by three test sets. In addition to performance metrics, we also generate the visualization results by t-SNE, which show PhosVarDeep can transform protein sequences to meaningful representations with strong discriminant power for phospho-variant prediction. The key contributions of our work are summarized as follows: (1) we exploit a Siamese-like deep neural network architecture with two identical subnetworks and a prediction module for phospho-variant prediction, which allows us to learn deep features on a pair of reference and variant protein sequences of each phospho-variant jointly, (2) we employ a deep neural feature encoding network PhosFEN in each subnetwork to capture sequence features related to phosphorylation by transfer learning, (3) we design the CNN module with two parallel multi-layer CNNs to learn variant-aware phosphorylation sequence features that are useful for phospho-variant prediction, (4) by effectively integrating the outputs of the above architecture with the prediction

**Table 7  Prediction scores of confirmed phospho-variants.**

| Gene | Protein | Variant | Phos.site | PhosVarDeep | MIMP |
|------|---------|---------|-----------|-------------|------|
| TP53 | P04637 | P47S | S46 | 0.943 | 0.743 |
| TP53 | P04637 | R213Q | S215 | 0.983 | 0.867 |
| TP53 | P04637 | R282W | T284 | 0.987 | 0.879 |
| BDNF | P23560 | V66M | T62 | 0.911 | 0.839 |
| PER2 | O15055 | S662G | S662 | 0.982 | <0.5 |
| MeCP2 | P51608 | R306C | T308 | 0.983 | 0.884 |
| NKX3-1 | Q99801 | R52C | S48 | 0.988 | <0.5 |
| ABCB4 | P21439 | T34M | T34 | 0.787 | <0.5 |
| GLUT1 | P11166 | R223W | S226 | 0.976 | 0.979 |
| CLIP1 | P30622 | E1012K | S1009 | 0.977 | 0.952 |
| CTNNB1 | P35222 | S37C | S33 | 0.824 | 0.756 |
| CTNNB1 | P35222 | G34R | S47 | 0.996 | <0.5 |
| Cyclin D1 | P24385 | T286R | T286 | 0.947 | <0.5 |
| hOG1 | Q16539 | S326C | S326 | 0.991 | <0.5 |
| UBE3A | Q05086-3 | T485A | T485 | 0.038 | <0.5 |
| PLN | P26678 | R14C | S16 | 0.993 | 0.925 |
| MAF | O75444 | P59H | T58 | 0.988 | 0.986 |
| Gab1 | Q13480 | T387N | T387 | 0.977 | <0.5 |
| hERG1 | Q12809 | K897T | T897 | 0.985 | <0.5 |
| STAT1 | P42224 | L706S | Y701 | 0.972 | <0.5 |

module, our method achieves remarkable performance for classifying phospho-variants on comprehensive experiments.

Although PhosVarDeep has shown promising performance in predicting phospho-variants, there is still considerable room for further improvement. Firstly, some cellular context information (*e.g.,* protein-protein interactions) is also useful for predicting phospho-variants (*Patrick et al., 2017*), which can be further integrated with sequence information in our future work. Secondly, as the deep learning method is still a black-box lacking interpretability (*Ma et al., 2018*), our method faces great challenges in explaining meaningful biological processes. In the future work, we will modify the framework to make our model more interpretable and realizable by combing some other modules, such as attention mechanisms (*Mnih et al., 2014*). Thirdly, the unsupervised approach to distinguishing potential phospho-variants is a promising alternative to solve the problem about the absence of a suitably large positive training set, which can be further used in our future study. Finally, as an effective computational approach for exacting and combining reference and variant sequence information, PhosVarDeep can be further advanced and extended to other types of variant prediction tasks.

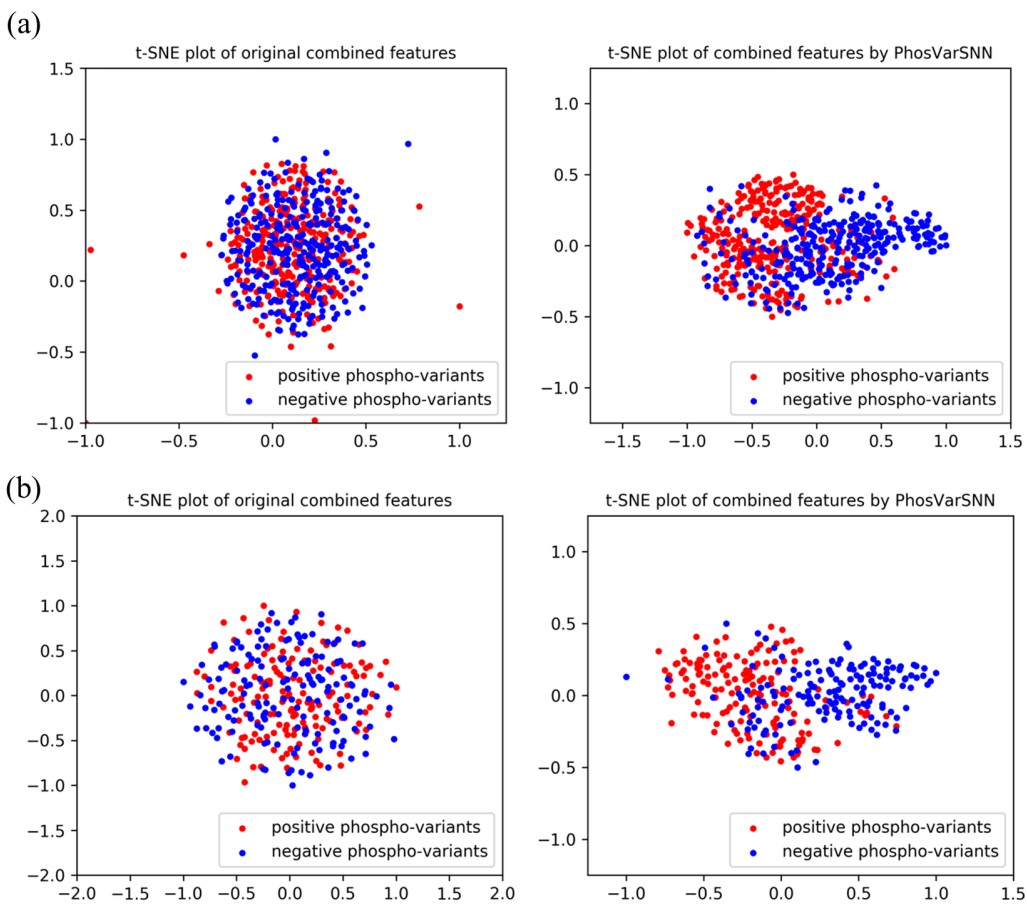

**Figure 4** **Visualization of original combined one-hot encoding features and combined features extracted by PhosVarDeep.** Red dots represent positive examples of phosphor-variants on (A) S/T sites or (B) Y sites of test set3, blue dots represent negative examples of phosphor-variants.

## CONCLUSIONS

In this paper, we propose a novel unified deep-learning framework named PhosVarDeep for accurate phospho-variant prediction. In order to efficiently extract and combine reference and variant sequence information, PhosVarDeep exploits a Siamese-like deep neural network architecture with two identical subnetworks for feature extraction and a prediction module for integrating the outputs of subnetworks. In each subnetwork, PhosFEN is employed to capture general phosphorylation sequence features by transfer learning, and a CNN module is designed to learn variant-aware phosphorylation sequence features, which helps to improve the performance of phospho-variant prediction. The experimental analysis on three phospho-variant test sets confirms the effectiveness of our proposed method, suggesting PhosVarDeep is a competitive and promising method in predicting phospho-variant and can provide clues for further biological research.

### Funding

This work was supported by the National Natural Science Foundation of China (No. 61871361, No. 61971393, No. 61471331, No. 61571414). The funders had no role in study design, data collection and analysis, decision to publish, or preparation of the manuscript.

### Grant Disclosures

The following grant information was disclosed by the authors:
National Natural Science Foundation of China: 61871361, 61971393, 61471331, 61571414.

### Competing Interests

The authors declare there are no competing interests.

### Author Contributions

- Xia Liu conceived and designed the experiments, performed the experiments, analyzed the data, prepared figures and/or tables, authored or reviewed drafts of the paper, and approved the final draft.
- Minghui Wang conceived and designed the experiments, authored or reviewed drafts of the paper, and approved the final draft.
- Ao Li analyzed the data, authored or reviewed drafts of the paper, and approved the final draft.

### Data Availability

The code is available at GitHub: https://github.com/lisalikegaga/PhosVarDeep. The data is available in the Supplemental Files.

### Supplemental Information

Supplemental information for this article can be found online at http://dx.doi.org/10.7717/peerj.12847#supplemental-information.

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
