# Peer review of "PhosVarDeep: deep-learning based prediction of phospho-variants using sequence information"

_PeerJ, doi:10.7717/peerj.12847_

## Round 0.1 · original submission · Major Revisions

While both reviewers are positive about the work, there are a number of concerns brought up by both, with aspects of the training and test data raising a number of questions. Please address these concerns as best as possible in your resubmission.

Reviewer 1 ·

Basic reporting

The article is generally well written. There are a few grammatical issues I noticed

In the title, ‘phospho-variant’ should be the plural ‘phospo-variants’

Line 16 ‘Researches’ should be ‘Researchers’

Line 30 ‘experiment’ should be ‘experimental’

Ensure when referencing authors in the text that the authors’ last name is used.

Experimental design

My main concern is with the positive training and test set used in this study. The authors claim they are taking positive examples of phospho-variants from PhosSNP. However, PhosSNP is not primarily a database of confirmed examples of phospho-variants (these are in the minority), rather it contains predicted phospho-variants, and predicted on the basis of sequence features. Given that the authors are also training a model on the basis of sequences features, this creates a system of circular logic, where the positive training and test sets have essentially been selected on the basis of sequence features and then trained and evaluated on sequence features. In this regard, it is not a surprise that they see such significant improvement in prediction accuracy over MIMP and PhosphoPICK-SNP.

In the absence of a suitably large positive training set, my question would be whether the authors could take an unsupervised approach to distinguishing potential phospho-variants? There should be plenty of putative pospho-variants available on the PhosphoSitePlus database as well as the high confidence negatives that the authors are already using.

There are similar issues with the evaluation of methods for use in the PhosFEN part of the model. The difference between the variant and reference site are being used to evaluate the ROC and AUC, but again, these sites have been chosen on the basis of GPS 2.0 predictions so I don’t understand how they can be considered a meaningful test set. As there are dedicated databases of experimentally confirmed phosphorylation sites (e.g. PhosphoSitePlus), surely it would make more sense to make use of these when evaluating what phosphorylation site predictor to use in PhosFEN?

Validity of the findings

As explained in section 3. I feel that the validity of the findings are undermined by an inappropriate positive training and test set.

Reviewer 2 ·

Basic reporting

In the paper “PhosVarDeep: deep-learning based prediction of phospho-variant using sequence information”, the authors proposed a deep-learning based prediction model to predict for phosphor-variant using paired reference and variant sequences as input. In their proposed model, the general phosphorylation sequence features were extracted by an existing pre-trained deep-learning model called DeepPhos. Then a CNN model was introduced to integrate the outputs of the pre-trained model. Finally, a prediction module is used for the final prediction of phosphor-variant. The manuscript was well organized and easy to follow. However, my major concern about this paper is the dataset they used and the performance evaluation. Is there any newer data that can be used for training and evaluation?
Major concern:
1 The authors should clarify the definition of phosphor-variants in their paper. As in the PhosSNP paper, they defined five different types of phosphorylation-related SNPs. But in this manuscript, they didn’t interpret the term of “phosphor-variant” and which type is their target.
2 How they collected the training data is not clear. The authors mentioned that they collected positive examples in PhosSNP and negative sets by Ralph et al. They should introduce these data and how they were collected. Are there any homology sequences between the positive and negative sets?
The phosSNP database also contains predicted phosphor-variants, they should also clarify that.
3 The database phosSNP was published 10 years ago, is there any newer data that can be used for the model training? I know another one published in 2019 in the paper “AWESOME: a database of SNPs that affect protein post-translational modifications”. But I don’t know whether they provide phosphor-variant based on experimental data.
3 One big concern of deep-learning-based methods is the overfitting problem. More complex models tend to obtain better performance in the training data than simple models. In line 124, what does the “strictly non-overlapping training, validation and test sets” mean? Does it mean no exact the same sequences in the training, validation, and test set? If so, it is not enough for biological sequence predictions. A standard approach is to remove similar sequences (>40% sequence similarity) between the training and test set.

Experimental design

no comment

Validity of the findings

More recent data is expected to evaluate the performance. The positive data used in this work was published 10 years ago.

---

## Round 0.2 · Minor Revisions

Thank you for addressing the reviewer concerns. There is only a minor comment from a reviewer suggesting the clarification regarding potential phospho-variants to avoid confusion. This seems reasonable, though this is at the authors' discretion. As the decision is "minor revisions," the manuscript will not require further reviewer input and can proceed to acceptance following the authors' review of the comments. Thank you again for your thoughtful revisions.

Reviewer 1 ·

Basic reporting

No comment

Experimental design

I appreciate that the authors have improved the definitions and description of the datasets used. Given that the ‘positive’ examples of phosphovariants that the authors are using are in fact phosphorylation sites in the vicinity of variants, rather than confirmed phosphovariants, it would be best to make clear that the positive training set is in fact comprised of potential phosphovariants rather than ‘positive examples of phospho-variants’. I appreciate that in the updated text, the authors refer to potential phospho-variants, however, in the first line of the introduction to the datasets for example, the training set is still referred to as positive examples, which could cause confusion to the reader.

Validity of the findings

No comment

Reviewer 2 ·

Basic reporting

The authors have addressed all my concerns in the revised manuscript. I have no further comments.

Experimental design

no comment

Validity of the findings

no comment

---

## Round 0.3 · accepted · Accept

Thank you for addressing reviewer concerns and congratulations again.